# Natural Capital Accounting Informing Water Management Policies in Europe

**Ioannis Souliotis and Nikolaos Voulvoulis \***

Centre for Environmental Policy, Imperial College London, London SW7 1NE, UK;
i.souliotis16@imperial.ac.uk
**\*** Correspondence: n.voulvoulis@imperial.ac.uk

**Abstract:** In the European Union, the Water Framework Directive provides a roadmap for achieving good water status and sustainable water usage, and a framework for the information, types of analysis, and interventions required by the Member States. Lack of previous knowledge in, and understanding of, interdisciplinary approaches across European countries has led to applications of corrective measures that have yielded less than favourable results. The natural capital paradigm, the assessment and monitoring of the value of natural capital, has the potential to convey information on the use of water resources and improve the connection between implemented measures and changes in the status of the resources, thus enhancing the effectiveness of policy interventions. In this paper, we present the natural capital accounting methodology, adapted to the requirements of the Directive, and demonstrate its application in two European catchments. Using economic methods, the asset value of two ecosystem services was estimated and associated with changes in water status due to policy instruments. Findings demonstrate that the asset value of water for residential consumption and recreational purposes fluctuates from year to year, influenced by current and future uses. Consequently, managing authorities should consider both current and emerging pressures when designing interventions to manage water resource sustainably.

**Keywords:** natural capital; ecosystem services; water framework directive; water management

## 1. Introduction

The natural environment is consistently undervalued in decision-making. However, besides the inherent value of natural resources to human wellbeing, a wide range of government policies including investments in infrastructure and economic growth are influenced by the value of natural resources and their availability [1]. Indeed, it is now increasingly recognized that environmental degradation diminishes the capacity of the planet to sustain economic development [2–4].

The presence of human pressures on water resources coupled with ineffective and unstainable management practices deeply affect the ability of ecosystems to deliver services. Ecosystem services are the source of benefits which people gain from natural ecosystems, and natural capital is the stock of natural ecosystems from which these benefits flow [5]. Reduction in the delivery or loss of ecosystem services results in economic losses, which, given the current monitoring schemes in Europe, are hardly considered by national economic policies. However, maintaining natural capital, i.e., ecosystems and their services, is fundamental to human welfare and development. Given the pressures and threats on European ecosystems, Europe risks losing natural capital without valuing what is being lost [6]. Methods of monitoring and assessing the importance of such services to a society and its economy have increasingly gained the

interest of governments in the last two decades [7], making the case for environmental protection.

The publication of the Millennium Ecosystem Assessment [8] ignited a broad discussion on the interaction of humans and the environment and influenced the development of assessment methods [9–15], providing the conditions for the development of approaches of natural capital accounting and assessments, a promising avenue for improving the status of ecosystems, while supporting policy making. Natural capital and ecosystem services are both definitions included in the ecosystem approach [16]. Natural capital, a term introduced by Pearce et al. [17], comprises the ecosystem and abiotic assets of earth that provide ecosystem services such as food, climate regulation, and recreation [18], or, as Costanza et al. [19] put it, natural capital can be described as "the stock of materials or information" contained within an ecosystem. Natural capital as a stock, provides flows of materials, energy, and information in the form of ecosystem services that, when combined with other forms of capital (social, human and or built capital), contribute to human welfare [20]. In other words, ecosystem services are the results of the interaction of biotic and abiotic components of natural capital [21] (Figure 1).

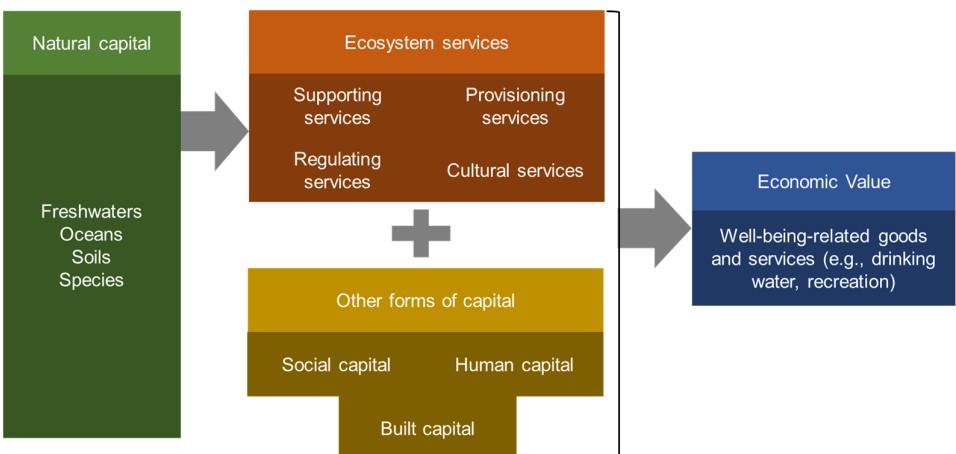

**Figure 1.** Natural capital assets, ecosystem services, and well-being.

Policy making is fundamentally concerned with choosing among various options (or combinations of different types of capital) to obtain the most valuable outcome. Consequently, valuation is an integrated part of designing and implementing policies. In this paper, we consider that the value of natural capital assets acts as an integrated indicator of the condition of the overall system (social and environmental). Increases in their value may also indicate an enhancement of the condition of natural capital or an increase in the marginal value of benefits provided to humans by the environment. For the purpose of monitoring the contribution of nature to welfare, natural capital accounting potentially has multifaceted roles in policy. Estimating the quantity and assessing the quality of natural capital assets systematically, as well as the benefits they provide to the economy and society, reveals how the use of resources influences economic development, thus providing opportunities for increasing the efficient use of natural resources, as well as for their protection [22,23]. The identification of pressures and possible risks provides the basis for an evaluation of the effectiveness of policy instruments, and fosters the adoption of practices that promote sustainability [24]. Furthermore, assessing how natural capital is affected by different industries of the economy has the potential to minimize emerging risks faced by businesses [25]. This is particularly important, as environmental issues take top spots in the World Economic Forum's Global Risks Report [26].

In this respect, a major advancement has been the development of the Environmental-Economic Accounting [27], led by the UN Statistical Commission with the involvement of international organizations such as the European Commission, the World

Banks, and hundreds of scientists and nongovernmental organizations. Since then, 24 countries, some of them in Europe (e.g., the United Kingdom, the Netherlands, Norway, Italy, Spain, Australia, and Canada) have compiled such accounts [28]. The accounting of natural capital aims at establishing consistent approaches of identifying, assessing, and monitoring the flow of goods and services and, consequently, the benefits generated by nature [29]. Overall, the natural capital principles and methodologies provide several important tools to managing authorities [30]. The use of a commonly accepted classification of ecosystem services and the identification, as well as recognition, of benefited stakeholders helps to organize information and frame each given management problem in a concise way. Following a standardized methodology to assess the value of ecosystem services and natural stocks also assists in keeping track of changes that occur over time.

In the European Union, the concept of natural capital accounting has been recognized by the EU Biodiversity Strategy to 2020 [31] and the Seventh Environment Action Programme of the EU [32], which highlight the importance of developing standardized natural capital accounting practices as a means to protect and enhance natural capital [33,34]. Additionally, the Eighth Environment Action Programme of the EU [33], which is to be adopted in 2021, prioritizes among other things the development and application of ecosystem-based management practices, including natural capital accounting and nature-based solutions [33]. Mainstreaming natural capital accounting for the implementation of environmental policy has therefore been an issue of increasing interest in the European Union, as it informs policymaking and fosters the implementation of nature-based solutions, which have the potential to provide higher socioeconomic benefits at lower costs compared to traditional approaches [35]. However, the concept is in its infancy; therefore, when the European Environment Agency implemented pilot projects [36], such as a project in the Warnow Basin in Germany, where different accounting applications were performed using WFD reporting data, it was concluded that the data sets that were contemporaneously available in Europe did not match the requirements of ecosystem accounting [18].

The WFD has been the main driver for the collection of data since its adoption in 2000. According to its provisions, Member States are required to develop River Basin Management Plans, which include an abundance of information [37] ranging from biological to socioeconomic data at the catchment level, aiming to assess the pressures on, and status of, inland waters, and to develop programmes of measures to improve the overall health of such ecosystems [38]. The lack of common definitions and objectives [39,40] as well the knowledge deficit of Member States in applying integrated methodologies has resulted in overall underperformance of the Directive [41,42]. In other words, the implemented programmes of measures did not provide the desired results, leading to a questioning of the effectiveness of the Directive [43]. This gave rise to the exploration of how approaches based on ecosystem services can be applied to foster a higher degree of integration between pressures, impacts, and programmes of measures to improve water status overall [44].

Acknowledging the importance of the WFD and developments on natural capital accounting, the aim of the paper is to explore its potential to inform the selection of programmes of measures and to provide a concise way of assessing how implemented measures impact the use and value of natural resources through changes in their overall water status. After a brief discussion on the connection between the WFD and natural capital, we present possible steps that could be followed to assess how policy interventions affect the value of natural capital, both through a theoretical and a practical approach. Finally, we apply natural capital accounting to estimate the asset value of two of the ecosystem services provided by water bodies in two case studies in Europe that have not yet used the information of such accounts for the development of River Basin Management Plans and the assessment of programmes of measures. Our aim is to test

how the natural capital approach can complement the implementation of the WFD to manage water resources sustainably.

## 2. Natural Capital and the WFD

The adoption of the WFD has been a decisive turn in water management in Europe [42]. Acting as an overarching legal document, the Directive aimed at embracing all fragmented pieces of water law in Europe, with the ultimate goals of preventing the deterioration of the quality of waters and achieving good water status by managing water re-sources effectively [45]. More importantly, the WFD introduced a new paradigm in water management by promoting integrated river basin management and stakeholders' participation, focusing on enhancing the overall health of the system instead of just the chemical status of water, and by including economic principles and tools as key features of its implementation. Furthermore, the Directive took up a systemic approach by introducing river basins as the main governance unit, therefore recognizing that each river basin constitutes an interconnected system [46,47]. Instead of managing specific elements in isolation of the broader system they are traced, the WFD took a decisive step away from the command-and-control practices introduced by traditional water management policies [48]. Additionally, compared to previous environmental Directives, the WFD set a specific date for achieving its objectives and requires the introduction of specific policy interventions, considering their cost-effectiveness [49]. Finally, the WFD requires the interventions designed and implemented by the Member States in each River Basin, as well as detailed information on the status of water resources, the types of pressures, water uses, socioeconomic characteristics, etc., to be included in the River Basin Management Plans (RBMP), which should be updated in fixed intervals (management cycles).Though promising, the implementation of the Directive has faced significant obstacles, leading to growing concern that many EU Member States will be far from achieving the objective of good status by 2027 [37]. According to the WFD fitness check [35] published in 2019, there had not been substantial improvement in the status of water in the first two cycles. Potentially, this is due to delays in the implementation of the Directive, the high number of deadline extensions that were granted to 40% of all surface water bodies and 11% of groundwater bodies [50], as well as misunderstandings about the definition of ecological status [51].

Deepening the understanding of the relationship between environment and society, two components of the same system, through the identification and assessment of the value of nature to humans, has been considered to improve catchment management [52]. Ecosystem services, the benefits that the environment provides to humans, has been suggested as a possible tool to shed light on the interaction between the two components of the socio-environmental system and to promote the protection and restoration of ecosystems [53]. As far as this paradigm is concerned, ecosystem services are the nexus between the condition of ecosystems and well-being, as the status affects the delivery of ecosystem services [54]. Several authors have used ecosystem services to demonstrate their suitability for the implementation of the WFD, for the purposes of economic analysis, design and implementation of programmes of measures, assessment of pressures, and stakeholders' participation [11,44,55–61].

The WFD does not refer explicitly to natural capital. Its purpose is to protect waters (inland surface waters, transitional waters, coastal waters, and groundwater), enhance their status, and promote their sustainable use through implementing PoMs that eliminate pressures and recover costs of water services [61]. However, if it is not technically feasible for a Member State to achieve a good status within the set timeframe, or if natural conditions do not allow for the achieving of a good status, or if costs are disproportionate to the benefits of improving water statues, extension of the deadline for reaching good ecological status or setting lower targets may be allowed [62]. The disproportionality principles apply when the financial ability of Member States is such that does not allow for the implementation of programmes of measure, or when the undertaking costs of

implementing measures are significantly higher than the benefits of improving water status [63,64]. In economics, however, disproportionate cost is not a standard concept [65], and there is no standards on which a benefit–cost ratio should be considered prohibitive. Moreover, the WATECO Guidance Document [66] suggests the use of economic tools to assess the disproportionality of costs, however it states that decisions on the need for derogation remain political. From an economic perspective, Cost–Benefit Analysis (CBA) is the obvious tool used to assess the disproportionality of costs. CBA considers the welfare value of benefits accruing from a change in the circumstances and compares it to the cost of policy options. On the contrary, natural capital accounting considers exchange prices of ecosystem services based on current pricing mechanisms and market conditions [28]. In cases where exchange prices cannot be obtained, it might be feasible to use welfare values, assumed as exchange values [67].

Natural capital accounting provides information on the condition of the ecosystem, the physical and monetary flow of ecosystem services, and the monetary value of ecosystem assets, therefore it constitutes a tool to measure the changes in the stock of natural capital. The process of designing programmes of measures and consequently assessing their cost-effectiveness and proportionality can be informed by natural capital accounts in the following ways (Figure 2):

1.  By identifying the users and uses of water resources within each catchment area that will be impacted the most by the policy intervention;
2.  By assessing the trade-offs between different ecosystem uses;
3.  By establishing a common currency to allow for a comparison of changes within and between each asset of each ecosystem;
4.  By incorporating information from a natural capital assessment into a CBA or other appraisal techniques.

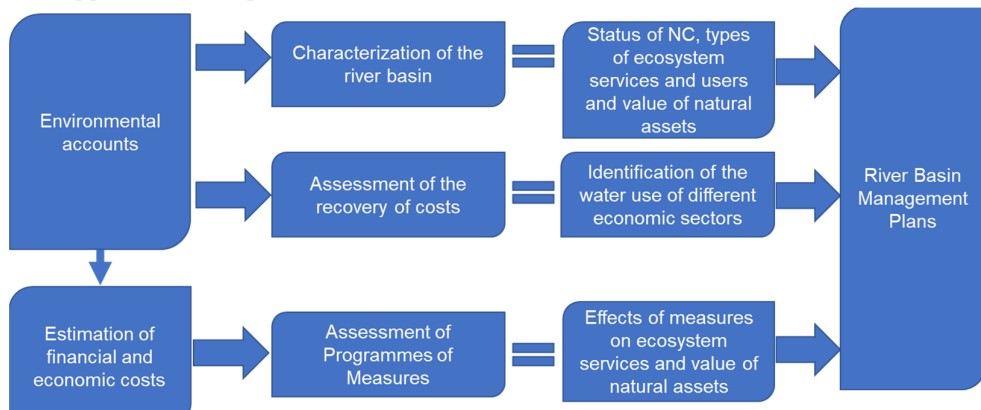

**Figure 2.** Use of natural capital accounts information for the implementation of the WFD.

### 3. Assessing the Value of Natural Assets in line with the WFD

Assessing the effectiveness of programmes of measures has been a troublesome experience for most of the EU Member States. However, from the First Implementation Report published in 2007 to the Fifth published in 2019, Member States have made significant progress concerning the development, assessment, and implementation of PoMs, although significant gaps still remain in translating the results of the economic analysis into concrete measures [68]. Meeting the targets of the WFD requires an increased investment in technical and non-technical measures, which will require sophisticated economic justification to facilitate water-related decisions. Estimating the stock value of the flow of services according to natural capital principles can be aligned with the required economic underpinning to better serve the needs of the Water Framework Directive. Taking that into account, this section describes the steps (Figure 3) that need to be taken

to obtain information on the status and contribution of the ecosystem and how this can be fed into assessments of policies.

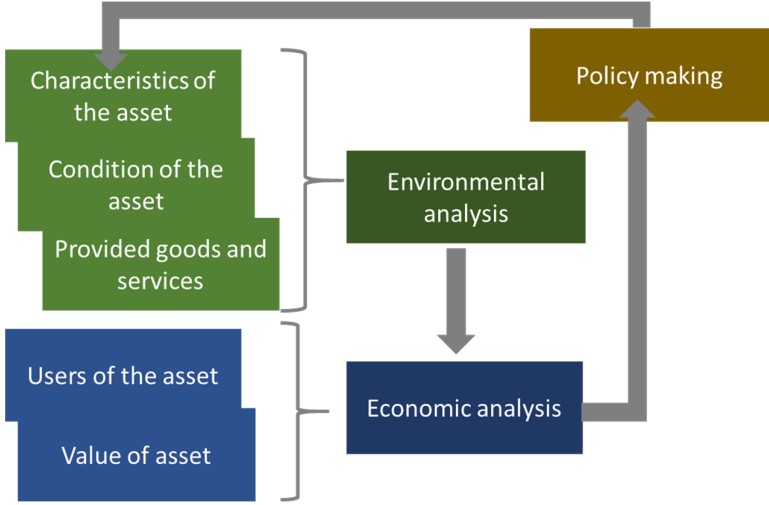

**Figure 3.** Steps for assessing the value of natural assets.

**Step 1: Characteristics of the water body.**

The initial step is to understand the components of the broader system encompassing the natural resource of interest which may influence policy outcomes. Therefore, general characteristics of the natural resource (asset) and the wider system help in constructing a baseline that considers land cover the classes or the type of the ecosystem and their extent. Such data may be spatial information, land use data, climate information, as well as information on the socioeconomic characteristics of the wider area that can provide an indication of current and future pressures. For this purpose, classifications of ecosystem assets may be used, such as the UK Broad habitat types [69]. Besides that, Article 5 and Annex VII of the WFD require policy makers to undertake an analysis of the characteristics of each River Basin District, a review of the impact of human activity on the status of water and an analysis of its uses for the drafting of the River Basin Management Plans.

**Step 2: Condition of the asset.**

Assessing the condition of the asset requires to consider the physical, chemical, and biological aspects of the resource. In Europe, the WFD provides detailed consideration to the meaning of good ecological status. More specifically, Annex V [70] set out a list of biological, hydromoprhological and physicochemical quality elements [71]. In addition to that, understanding the relationship between the characteristics of the aquatic ecosystem and human pressures helps to design targeted measures to improve good ecological status of water systems [55].

**Step 3: Types of goods and services the asset provides to water users.**

Aquatic ecosystems provide a wide range of critical ecosystem services that can be categorized into provisioning (e.g., water provisioning and fish production), regulating, supporting and cultural services (e.g., recreation) [19,72–74]. Identifying the specific ecosystem services provided by the natural capital, their flow and the users that benefit from these services is an essential part for obtaining a preliminary indication of the importance of natural capital. Additionally, it helps determining the direct and indirect benefits to users, option value and the related non-use values (existence and bequest) [75,76].

**Step 4: Value of the provided goods and services.**

Economic value for stream of services relates to the contribution of ecosystem services to human welfare and broadly speaking is measured based on each individual's own preference and assessment on their wellbeing [77]. Costanza [20] proposes three different paradigms of assessing the value of natural capital. The first relates to "Homo economicus", where value is obtained through individuals' stated or revealed willingness to pay; the second relates to "homo communicus", where the community rather than the individual [78] define the value of natural capital; lastly the third is associated with "homo naturalis". According to it, individuals are integrated components of the system, therefore value encompasses social, biophysical and economic dimensions of the ecosystem services [79,80]. Nevertheless, in the natural capital context, emphasis is given to the value of past, current, and future flow of benefits of ecosystem services. The flow of benefits are discounted to present values to estimate the total benefit of an environmental asset [81]. At the EU level, the methods used for valuing ecosystem services depend on the goal served by each particular account [22]. Furthermore, the SEEA EEA classifies the valuation methods in three broad categories: market based or cost-based methods (e.g., unit resource rent, production function, replacement cost, defensive expenditure, averting behaviour), revealed preference methods (e.g., hedonic pricing and marginal values from travel cost demand functions) and stated preference methods (e.g., contingent valuation and choice experiment).

As policy makers need to evaluate the effects of policy changes the value of flows of benefits could play an important role in the assessment of management options. An advantage of the natural capital accounts is that they include not only the economic value of ecosystem services, but also physical data on the natural capital stock. This is particularly important when policy makers from various organisations, need to implement integrated methodologies, such that proposed by the WFD. The prime focus of natural capital accounts is to reveal the ecosystems' contribution to the economy [28]. Additionally, natural capital accounts can be used either for backward-looking or forward-looking assessments. For instance, assessment and monitoring of environmental-economic macro-indicators, reviews on implemented projects concerning expenditures and benefits and sustainable development monitoring, national development plans, land use strategic planning to name a few [1]. Vardon et al. [82] explain that information dwelling from natural capital accounts can inform decision makers at any stage in the policy cycle (agenda setting, policy implementation and evaluation and measuring success). Natural capital accounting can be used in parallel to other economic methods, such as Cost–Benefit Analysis (CBA) suggested by WFD supporting documents (e.g., WATECO 2003). While CBA considers the flows of services and their benefits, natural capital accounting considers the stocks of natural resources, thus incorporates sustainability considerations that cannot be captured by CBA [23].

## 4. Materials and Methods

### 4.1. Description of Case Studies

Two case studies in Europe, one in Greece and one in the UK were selected for applying the natural capital approach. Both areas are operational catchments within a River Basin District and were studied by the GLOBAQUA (Grant agreement no. 603629-ENV-2013-6.2.1-Globaqua) project [83]. To our knowledge, natural capital accounts were not used in the development of River Basin Management Plants in either of the two countries, indicating that if not at all only to a limited extent the stock value of ecosystem services affected policy decisions. The UK has undertaken a national ecosystem assessment [84] and since 2013 the Office of National Statistics has been publishing annual environmental and ecosystem accounts [85]. Data on natural capital accounts are available through the Department for Environment, Food and Rural Affairs (Defra) and the UK Environment Agency and the Office for National Statistics and can be easily accessed by

the public. On the contrary, Greece has not yet compiled and published natural capital accounts. The main portal for all environmental information is that of the Ministry of Environment and Energy, however datasets on EU environmental legislation are not available [86]. Data on aspects of the WFD can only be extracted from the River Basin Management Plans that have already been submitted to the European Commission and no background documents are accessible. Therefore, the selection of these case studies helps to explore the difficulty in using ecosystem-based approaches in more and less methodologically advanced countries. Our aim is to promote the development of accounts at minimum for some ecosystem services based on information that is already available from WFD reporting. The section starts with a general description of the areas, including the status of water resources, present pressures and socioeconomic characteristics.

The first, the Evrotas River Basin (RB) is located in the Eastern Peloponnese River Basin District in Greece in the Prefectures of Laconia and Arcadia (Figure 4). The catchment area occupies the biggest share of the basin, with a length of 93 km and total catchment area of 2410 km² [87]. The main tributaries are the Oinountes, Magoulitsa, Gerakaris, Kakaris, Rasina, Mariorema, Xerias [88]. Overall, there are 44 river systems in the Evrotas RB. The cli-mate of the area is Mediterranean with high levels of precipitation, however the low ratio of mean annual precipitation to potential evaporation characterizes the area as semiarid [89]. Furthermore, in the last 35 years decreasing trends in rainfalls and discharge have been observed [90].

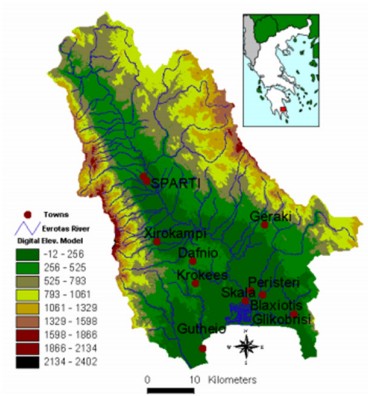

**Figure 4.** The Evrotas River basin [90].

According to the River Basin Management Plan of Eastern Peloponnese the main pressures in the Evrotas catchment are related to water quantity, water abstraction for irrigation and droughts. Additionally, pressures on the quality of the water relate to agricultural activities (e.g., use of pesticides), aquaculture/fish farming, urban waste, septic tanks, and mining. Humans intervene in the area by removing natural vegetation, constructing embankments and by removing riverbed material leading to morphological pressures.

The second, the Broadland Rivers catchment covers an area of 3200 km² and it is mostly rural. The catchment includes 94 river water bodies with the four main (sub-catchments) being the Bure, Wensum, Yare and Waveney and 19 lake water bodies [91] (Figure 5).

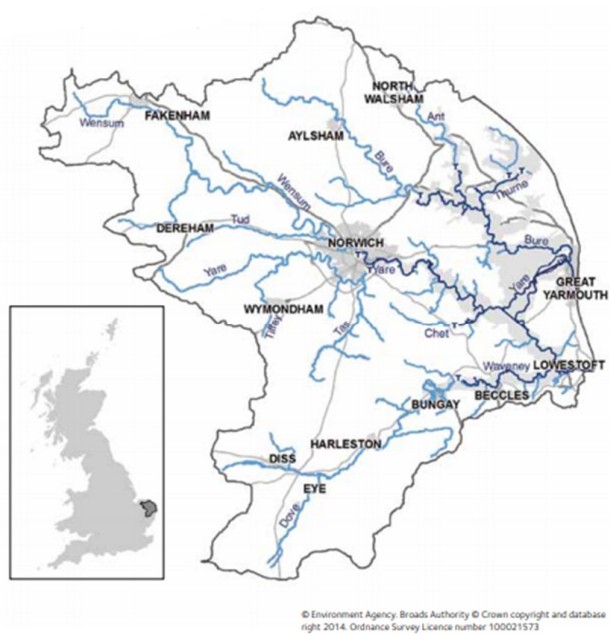

**Figure 5.** The Broadland Rivers catchment. Source: Environment Agency [92].

The largest settlements within the catchment area is the city of Norwich and the seaside towns of Great Yarmouth and Lowestoft. Additionally, the catchment encloses the Broads Executive area, which has the management status of a national park [92]. The vast majority of the area (approximately 87%) is used for agricultural purposes (non-irrigated arable land and pastures). Urban areas (including parks, industrial, commercial, transport units, mines, dump, and construction sites) account for 7.56% of the total area, while the remaining 3.92% is covered by forest and other nature units [48].

As far as the water status in the case studies is concerned (Figure 6), 70% of the rivers in Evrotas catchment achieved good status in 2017 compared to 32% in 2013 [93]. On the contrary, in Broadland Rivers catchment, the status of water bodies has progressively deteriorated. More specifically, while two rivers were at good status in 2015, none of them maintained the same status in 2019, where the majority of rivers were classified as moderate and the remaining as poor [94]. Giakoumis and Voulvoulis [48] claim that happened because the programme of measures developed by the management authorities focused on managing specific quality elements, thus neglected the connection between the pressures and the overall health of the system.

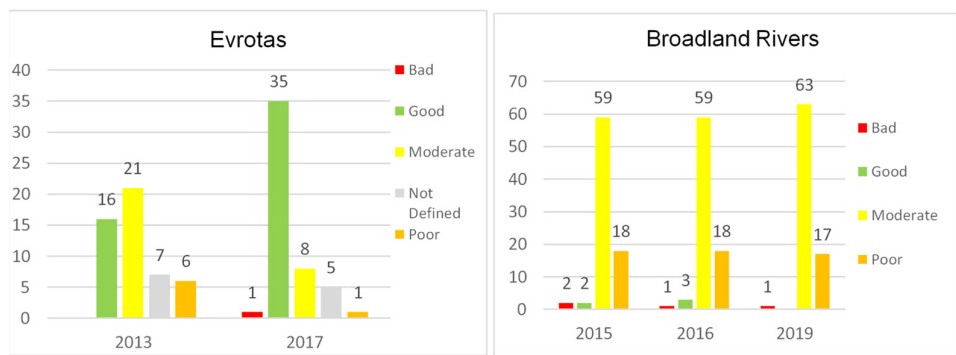

**Figure 6.** Water Status of Evrotas catchment in 2014 and 2017 and Broadland Rivers catchment between 2015 and 2019 (Information was taken from the River Basin Management Plans and

background documents, where available. The status for river bodies in the Evrotas catchment in 2017 consists of the projections reported to the European Commission).

In relation to pressures in the two areas, the most dominant in the Evrotas catchment relate to agricultural activities and concern mainly overexploitation of water resource for irrigation [95]. Overextraction results in partial desiccation in late summer-early autumn [96]. The catchment also faces diffuse agrochemical pollution and pollution from the wastewater treatment plant of Sparta [97]. Similarly, agricultural pressures exist in Broadland Rivers. Giakoumis and Voulvoulis [57] developed a framework that allows for the ecosystem approach to be operationalized for assessing and ranking pressures based on stakeholders' perception. They concluded that the most prominent pressures in the Broadland Rivers catchment are intensive nutrient or pesticide use, activities related to agricultural enhancement, pollution from urban areas and abstractions for potable supply.

*4.2. Collection of Data*

For the estimation of the asset value of ecosystem services a number of data sources have been used. Where possible data was taken directly from the River Basin Management Plans. In cases, where additional information was needed, official national data sources were accessed. Overall, information on abstractions and water uses is included, as well as data on visits to the catchment area for recreational purposes. Value adjustments were performed when necessary given data from the Statistics Offices of each respective country. To overcome the issue of missing data and in order to present comparable values we used proxies. For example, for the estimation of value of recreation number of overnights visits and number of trips were considered in the Greek and the UK case respectively. The lack of adequate information resulted in a limited assessment of the benefits of flow of ecosystem services. Overall, data processed for the different types of analysis is believed to be reliable as taken directly from official bases. Nevertheless, the following sections include a detailed description of the data sources.

*4.3. Estimation of the Value of Natural Capital*

In the subsequent sections, the asset value of water for residential purposes and the value of recreation of Evrotas and Broadland Rivers basin is estimated.

4.3.1. Water for Residential Use—Evrotas

Regional economic activity includes mainly agricultural, livestock and industrial activities. Agriculture is the main user of water. Public and private wells that cover the needs of the sector are estimated to be around 3500 [98], however there are a great number of illegal surface abstractions of surface water [90]. For this reason, only the residential water supply was considered. Water pricing in the River Basin area is differentiated into eight categories based on the type of user (public supply, agriculture, industry etc.).

In terms of residents, the Evrotas catchment area hosts approximately 70,000 permanent inhabitants [93]. Data concerning the population in the catchment area was obtained from the first River Basin Management Plan [99]. The document includes data for 2011 and projections for 2015 and 2021. Based on that we estimated the population for the missing years. For the estimation of water use per year, we followed the assumptions of the River Basin Management Plan (t Plan reports a unit value of output of 0.68 €/ m³. By taking into consideration costs related to compensation of employees and depreciation of capital, the unit cost of abstracting water was estimated to be approximately 0.218 €/ m³. Taxes and subsidies on water extraction were not included in the valuation as they are not relevant for the water supply in Greece). More specifically, it was assumed that each person consumes 250 liters per day. Based on that and the unit value of output taken directly from the River Basin Management Plan the total value of output was calculated. Additionally, by considering the amount of water abstracted by both the water and

sewage companies and the municipal utilities operating in the area and the capital and operating costs of these suppliers, the unit cost per $m^3$ of supplied water and consequently the total cost of supplied water were estimated. By subtracting the cost from the total value of output, resulted in the value of water for residential use for each consecutive year (Table 1).

**Table 1.** Value of water for residential use (£ million, 2019 prices).

| Year | 2011 | 2012 | 2013 | 2014 | 2015 | 2016 | 2017 |
|---|---|---|---|---|---|---|---|
| Flow (Mm³) | 6.3 | 6.3 | 6.4 | 6.5 | 6.5 | 6.6 | 6.7 |
| Value | 2.6 | 2.8 | 2.7 | 2.6 | 2.5 | 2.5 | 2.6 |

4.3.2. Water for Residential Use- Broadland Rivers

Estimating the annual value of water for residential used was based on information on the number of licenses and the maximum permitted volume per licenses derived from an Environment Agency database (Water_Abstractions_20150101.mdb). Using GIS data provided by the Environment Agency [100], a layer was included to obtain the licenses that are relevant for the Broadland Rivers catchment. As in the case of UK natural capital accounts [101] only water abstracted for public water supply was included in the estimation. Since the quantities associated with the water abstractions in Broadland Rivers considered the maximum abstracted quantity allowed, the used volume of water was calculated for each given year by computing the volume of used water as a percentage of the maximum abstracted quantity allowed included in the national water accounts for the England region (Table 2). Since input-output tables publicly available by the Office for National Statistics were referring to the national level, we used the annual values provided for England, which were then adjusted to calculate the value of flows of this service for the Broadland Rivers Catchment.

**Table 2.** Value of water for residential use (£ million, 2019 prices).

| Year | 2011 | 2012 | 2013 | 2014 | 2015 | 2016 | 2017 |
|---|---|---|---|---|---|---|---|
| Annual flow (Mm³) | 71.10 | 71.41 | 72.30 | 76.27 | 74.97 | 77.83 | 148.67 |
| Annual value | 26.13 | 30.84 | 30.19 | 25.46 | 18.92 | 41.73 | 57.39 |

4.3.3. Recreation-Evrotas

The estimation of the annual value of recreation in the case of Evrotas was a troublesome task due to lack of data. The revised RBMP includes information on the number of overnight stays for 2005 to 2009 and estimated number of overnights stays for 2015 and 2021. Based on the annual percentage change of this variable, the missing values for 2011–2014 and 2017 were estimated. Additionally, the number of trips was calculated based on the average duration of stay in days (7.35 overnight stays per trip) obtained from the Institute of the Association of Greek Tourist Enterprises [101]. Finally, the value transfer approach [102] was used to estimate the value per trip. More specifically, Latinopoulos [103] used the travel cost method to assess the demand for outdoor recreational services for protected riparian ecosystem in Northern Greece and it was considered the most relevant to the catchment of Evrotas. By adjusting for inflation, the results of their study show an average consumer surplus value of 217.7 (€, 2017 prices) per trip was obtained. Multiplying this value by the estimated number of trips yielded the total annual value of recreation (Table 3).

**Table 3.** Estimated value of recreation (£ million, 2019 prices).

| Year | 2011 | 2012 | 2013 | 2014 | 2015 | 2016 | 2017 |
|---|---|---|---|---|---|---|---|
| Overnight stays | 184,100 | 188,850 | 193,723 | 198,721 | 203,100 | 203,367 | 203,634 |

| Estimated num-ber of trips | 25,048 | 25,694 | 26,357 | 27,037 | 27,633 | 27,669 | 27,705 |
|---|---|---|---|---|---|---|---|
| Annual value | 4.79 | 4.91 | 5.04 | 5.17 | 5.29 | 5.29 | 5.3 |

### 4.3.4. Recreation-Broadland Rivers

The value of recreational benefits related to the Broadland Rivers basin was estimated using the Travel Cost Method (TCM). The method which suggested by Hotelling [104] and fully developed by Clawson [105], suggests that the recreational benefits at specific site can be estimated based on a demand function that accounts the number of trips/ number of visitors to the actual costs of a given visit [106,107]. Generally, the recreational demand function can take the following form [103]:

$$V_i = f_{(c_i, cv_i, sc_i, q\ )},$$

where $V_i$ denotes the number of visits, $c_i$ the cost of the visit, $cv_i$ the characteristics of the visitor, $sc_i$ the characteristics of the site and $q$ other parameters that may relate to the visit.

Traditionally, on-site surveys are implemented followed by econometric analysis. For the purpose of our study, the main data source for the analysis was the Monitor of Engagement with Natural Environment (MENE) Survey [108] dataset. MENE datasets include a wide range of information related to the visit location, travel and visit time, costs related to visits as well as the socioeconomic characteristics of the visitors and are considered an appropriate source for ecosystem accounts [109]. The study considers data for 2011 to 2019 for Suffolk and Norfolk counties, as case study area lies within these two locations. Furthermore, only responses that relate to visits in rivers, lakes and canals are considered (the datasets include visits to other locations such as mountains, woodland, beaches and parks among others).

Given that the working status of individuals included people in the labour market, unemployed individuals, pensioners, and students, in order to estimate the opportunity cost of each visits the hourly paid wage of each year was taken into account (Annual Survey of Hours and Earnings datasets for 2009 to 2019), which was then adjusted to 2019 prices using the current prices index (CPI). Concerning the cost of travelling, the respondents mentioned several modes of transportation (e.g., bike, car, train, bus, foot etc.). Besides those that travel to the location by bicycle or on foot, the cost of travelling was estimated as follows. For those that travelled by car both the travel distance as well as the average cost per mile was considered. The later was taken from RAC reports [110–112] for 2011 to 2013. For the remaining years the values were adjusted for inflation. Self-reported expenditure on bus and train fares was extracted from the MEME database. In cases where no cost was reported, the average cost per mile was estimated by the responses of other individuals for each given year. As for as the cost of travelling by taxi is concerned, data on tariffs were taken from annual Taxi Fares and Tariff consultation reports [113–115]. Expenditures for visitors travelling on foot and by bicycle were considered negligible. Concerning the cost of travelling time, the average speed of car in England [116] as well as the average speed of train [117] were considered. Based on that as well as the self-reported distance from the starting point of each respondent's trip, the travelling time was estimated given different modes of transportation. Additionally, this time was multiplied by 75% of the average hourly wage, as suggested by Fezzi et al. [118] to estimate the opportunity cost of time spent on travelling. Finally, expenditures on entrance fees and consumables goods on sites were taken directly by the MEME dataset.

For the estimation of the value per trip, two models were considered namely a Poisson regression model and a Negative Binomial model. This was due to the fact that the number of visits, which was the dependent variable in the model is a nonnegative integer and the frequency of small numbers of visits consisted a sizable fraction of the data set [119]. Dependent variables that were used for the estimation besides the cost per visit

were the age of the respondent, the work status, whether they own a car or not and the size of the household, however only the cost per visit variable was significant. Therefore, the other variables were finally omitted. As shown below (Table 4), in both models the cost per visit coefficient as expected carries a negative sign. For the used models the surplus of the individual *n* estimated as $S_n = \lambda_n / -\beta_{tc_r}$, where $\lambda_n$ is the expected number of trips (here 1) and $\beta_{tc_r}$, the estimated coefficient of the cost per trip variable. The estimated values were £ 69.32 for the Negative Binomial model and £ 58.01 for the Poisson model.

**Table 4.** Estimates from the Negative Binomial and Poisson model.

|  | **Negative Binomial Model** | **Poisson Model** |
|---|---|---|
| **Variables** | | |
| Constant | 1.347 *** | 1.371 *** |
|  | (0.0666) | (0.0411) |
| Cost per visit | −0.014 *** | −0.017 *** |
|  | (0.0035) | (0.0028) |
|  | AIC: 989.27 | AIC: 1158.3 |
|  | 'log Lik.' −491.636 | 'log Lik.' −577.1431 |

*** coefficient significant at $p \leq 0.001$.

In order to obtain the total value of recreation for visitors, we calculated the percentage of individuals traveling to a location with a river within the catchment area. By considering the number of tourists in England (excluding business trips) from 2011-2019 [120–128] and obtaining the number of tourists in the Broadland Rivers Catchment in 2018 [129] (we estimated the percentage of the total tourists in England visiting the area of interest. In order to estimate the individuals traveling to the catchment in 2018, we made use of the Outdoor Recreation Valuation Tool (ORVal: Version 2.0) developed by the Land, Environment, Economics and Policy Institute (LEEP) at The University of Exeter. The tool consists of an interactive map that allows the selection of land covers at various scales and provides the corresponding economic values related to recreational benefits. For the current study all Middle layer Super Output Areas (MSOA) that correspond to the study area and contain rivers canals and lakes were selected, which allowed the estimation of the number of tourists making a visit to a water site. Keeping the above percentage constant, the number of tourists in Broadland Rivers was estimated for each year. Finally, by multiplying the number of tourists with the consumer surplus ($S_n$), the total value of recreation was estimated (Table 5).

**Table 5.** Estimated value of recreation 2011–2019 (£ million, 2019 prices).

|  | 2011 | 2012 | 2013 | 2014 | 2015 | 2016 | 2017 | 2018 | 2019 |
|---|---|---|---|---|---|---|---|---|---|
| Number of visitors (thousands) | 954 | 725 | 900 | 700 | 583 | 1053 | 690 | 594 | 940 |
| Annual value (Poisson model) | 55.3 | 42 | 52.2 | 40.6 | 33.8 | 61.1 | 40 | 34.4 | 54.5 |
| Annual value (Negative Binomial model) | 66.1 | 50.2 | 62.4 | 48.5 | 40.3 | 73 | 47.8 | 41.1 | 65.1 |

The final step includes the estimation of the asset value of ecosystem services. This was done by estimating the net present value of future flows of the ecosystem services benefits [130,131]. As there is not an expected life span for the two catchments, the flow of future value was estimated using a 100-year asset life as indicated by the UK Office of National Statistics [132]. Furthermore, concerning the discount rate, estimates assume a

3.5% discount rate for estimated projected out to 30 years, declining to 3.0% up until 75 years and further declines to 2.5% thereafter.

## 5. Results

By completing the procedure described above, the asset value of the two ecosystem services was obtained. The results indicate how past and current water management practices adopted to eliminate pressures and improve the status of water, influence the economic value of harvested ecosystem services. Accounts such the following that integrate information on the economic consequences of interventions in a systematic and rigorous manner can be expected to provide support for assessing the effectiveness of programmes of measures and the overall effectiveness of the Directive.

Table 6 presents the asset value of the two assessed ecosystem services. As observed the values demonstrate fluctuations across years, which is expected given differences in the intensity of use and in the unit value from year to year. For example, water abstraction in Evrotas obtains the highest value in 2011, it declines in 2012 and finally starts increasing again in 2016. As the population and the flow of water increases from year to year, the stock value of water follows the trend of the unit price. Nevertheless, in both Evrotas and Broadland Rivers, there is clearly an increasing trend from 2015 onwards. A way of interpreting this, is that from 2015 onwards the contribution of providing water to households becomes more dominant. If such a trend continued policy makers should be alarmed, as sudden events that may influence the availability of water may have a severe impact on the wellbeing of households. Comparing the two cases, it is noted that on average the annual change of the volume of water abstractions is higher in the Broadland Rivers than in Evrotas, which in the WFD context potentially signifies that the latter catchment faces milder pressure on water from residential consumption than the former. Concerning the asset value of water abstraction, as it increases steeper in the Broadland Rivers, it could be said that the dependence of the economy on this specific service is higher in the UK case study, which could mean that the pressures on water from residential consumption will most likely be more intense in the Broadland Rivers in the future.

**Table 6.** Estimated asset value (£ million, 2019 prices).

| | Catchment | 2011 | 2012 | 2013 | 2014 | 2015 | 2016 | 2017 | 2018 | 2019 |
|---|---|---|---|---|---|---|---|---|---|---|
| **Water for residential use** | **Evrotas** | 73 | 68 | 71 | 67 | 60 | 68 | 73 | | |
| | **Broadland Rivers** | 763 | 817 | 829 | 807 | 765 | 840 | 937 | | |
| **Recreation** | **Evrotas** | 136 | 128 | 135 | 130 | 118 | 134 | 144 | | |
| | **Broadland Rivers (Poisson model)** | 1491 | 1347 | 1379 | 1322 | 1263 | 1341 | 1300 | 1267 | 1306 |
| | **Broadland Rivers (Negative Binomial model)** | 1866 | 1609 | 1647 | 1580 | 1510 | 1602 | 1554 | 1514 | 1561 |

Furthermore, in Broadland Rivers, as previously discussed water quality declined from 2015 to 2019. During these years however, the asset value of water abstraction increased significantly leading to a peak in 2017 following a notable rise in the flow of abstracted water. These opposite effects may indicate that long-term household water consumption is unsustainable. This signals that the managing authorities may be required to adjust the WFD programme of measures or even develop new that will further disincentivize households for consuming excessively.

As far as the value of recreation is concerned it also exhibits volatility from year to year. By construction, this variable measures the amount of time people spend outdoors, thus the

changes can be attributed to that rather than the money people spend on recreation. In the Broadland Rivers case the estimates obtained through the use of the Negative Binomial model demonstrate similar trend to the values included in the UK Natural Capital Accounts [133]. Additionally, the asset value of the flow of recreation is higher than that of abstracted water in both cases, which indicates that the contribution of recreation to the economy is higher. For policy purposes, this proposes that programmes of measures developed in the context of WFD, could be such that are effective at improving water status, while improving the sustenance and provision of recreational ecosystem services.

## 6. Discussion

Overall, the two case studies differ in terms of improvements on the status of the rivers within the catchments. The RBMP for Eastern Peloponnese show that a high number of rivers in the Evrotas case reached good status in 2017. On the contrary, in the Broadland Rivers catchment most of the water bodies reached moderate status in 2019, as the condition of some water bodies deteriorated.

To our knowledge, neither of the two catchments has used the natural capital methodology in the process of developing the RBMP. However, the tables presented above can be fed into policy analysis to further improve the implementation of the WFD. More specifically, besides improving the description of the case studies, accounting tables can be used to inform the development of programmes of measures and can be utilized in the assessment of the recovery of costs. For example, the Evrotas case study might be more susceptible to pressures related to tourism than to water abstractions in the future, as the number of overnight stays has been increasing, whereas the volume of water abstractions remained almost constant throughout the years. From a policy perspective, this could impose either an opportunity or a threat for the sustainable use of water resources. On one hand, a higher number of tourists could mean that a higher share of the natural capital will be used for satisfying related needs. Consequently, the use of land in the area might change in the future, as there will be a higher demand for tangible and intangible amenities, such as public transportation and lodging facilities, accommodation facilities (e.g., hotels), parking places, and transportation facilities, bringing about the emergence of new pressures on water resources. On the other hand, policymakers could further improve the overall health of the ecosystem through the adoption of green measures, such as green streets, pocket parks, and tree planting [134,135] that play a critical role in protecting water resources and providing opportunities for recreation, among others things, and hence benefit society [136]. As a result, natural capital accounting can assist in identifying areas for public investments that can simultaneously promote human development and the conservation or restoration of natural capital [137].

On the contrary, the Broadland Rivers catchment seems to be more susceptible to pressures related to water abstractions for residential purposes than those related to recreation, given that, from 2016 to 2017, there has been a major increase in the amount of abstracted water. Future population increases could pose a threat to the sustainable use of water resources if water consumption is not adequately controlled. Nevertheless, the Water Exploitation Index [138] for the two countries demonstrates that Greece has been facing increasing pressure on renewable freshwater resources from 2015 onwards, whereas the overall position of the UK has relatively improved compared to previous years. However, the information concerning this index provided by the European Environment Agency refers to the national rather than the catchment scale, therefore disregards regional and seasonal changing conditions. Besides that, as far as the value of recreation in the Broadland Rivers is concerned, we observe that the average growth rate is negative, which might mean that management practices adopted in the area might have had a negative impact on the society. Information on potential effects of measures on ecosystem services provided by the Environment Agency [92] verify this result. More specifically, the cultural services of Waveney River (one of the main water bodies in the Broadland Rivers catchment) were expected to be negatively impacted by the proposed measures, as implemented measures could negatively influence areas and structures of cultural interest. Assuming that policy interventions that effectively target pressures are realized, the

status of water should be expected to further improve in the Broadland Rivers case, which will provide further opportunities for harvesting ecosystem services in the future. This, on one hand, could increase the annual flow of ecosystem services [139], however reductionist planning that does not account for the effects of measures on other aspects of the resource might lead to changes in the overall functioning of the socioenvironmental system, therefore leading to lower-than-expected benefits. Taking this into account, we suggest that the presented procedure can be of great use to managing authorities. By developing natural capital tables, managing authorities are enabled to obtain insight on the current use of natural resources, as well as the potential aspects that could be influenced in the future to design effective corrective measures.

In other words, such information can be used to monitor the development of the economic–environmental system and form the basis for evaluating the trajectory of future development and the effectiveness of programmes of measures. Natural capital accounts assess the stock value of natural capital and can signal whether PoMs contribute to sustainability. This is particularly important for appraisals of spending options, where considerations such as securing benefits for future generations need to be considered [23]. For the purposes of the WFD, such information can supplement cost–benefit analysis, which focuses on the flow of benefits from nature [140]. Besides this, through natural capital accounting, policymakers can evaluate the impact of measures on specific ecosystem services, identify the stakeholders that are affected by water status changes, and assess the unintended consequences of policy responses [141]. In addition to that, environmental indices created to measure the interaction of society to environmental resources [142] such as the water resource vulnerability index [143] can complement the natural capital accounting by deepening our understanding of ecosystem changes [144,145].

Finally, as natural capital accounting methodologies are recent developments [146], and because several issues concerning the contribution of natural capital to the economy are still to be resolved [147], caution should be taken when undertaking such an analysis and interpreting results. Some of the most dominant issues include uncertainty pertaining to our capacities to anticipate the future, the quality of gathered information, and a faulty understanding of the system of interest [148]. Nevertheless, developing standards for natural capital accounting and further improving current methodologies can foster a better understanding of the complexities of the system, transforming them into manageable risks through the use of a single unit of measurement to express the condition, extent, and value of different aspects of nature [149,150], thus improving water management.

## 7. Conclusions

The European Commission defines natural capital accounting as a tool with which to monitor changes in the stock and condition of natural capital at different scales and a means to integrate the value of ecosystem services into reporting systems [151]. As shown in earlier sections, developing natural capital accounts requires a great amount of data, such as detailed information on ecosystem services supply, assessment of the status of ecosystem assets, and an identification of the uses of the ecosystem services as well as their value. In this study, we presented the links between the steps of the implementation of the WFD and the development of natural capital accounts. Overall, monitoring annual changes in the state of an ecosystem is both a requirement of the WFD and a prerequisite for developing natural capital accounts [130,152]. Assessing trends in ecosystem services can increase our understanding of how the environment functions [153] and shed light on the dynamics of the interactions between societies and the environment.

Environmental accounts and, in particular, water accounts, have had many applications around the globe, from preparing catchment management plans and assessing the level of cost recovery [154,155] to monitoring progress towards sustainable development [156]. Assessing how the economic value of the services of interest and the status of water change across years provides a useful insight for policymaking can reveal the added value of investing in nature. As per the WFD, EU Member States are obliged to design and implement measures to prevent

further deterioration of the quality of waters and improve their overall status. The measures implemented in the Broadland Rivers to some extent failed to achieve that [48]. Therefore, the question arises as to whether traditional measures besides being able to improve water classification can deliver benefits to society. Natural capital accounts have the potential to contribute to the answering of this question, as the obtained economic value incorporates information about the structure of the institutional setting, the intensity of ecosystem services harvesting, and the extent and condition of natural resources [157,158]. They can provide information on trends across time and allow for comparability among river basins, measure effects of policy interventions on water resources, and give an indication of the cost-efficiency of policies aiming to improve the health of the environment [159]. Taking into account that there are still significant gaps in the assessment of PoMs [160], natural capital accounting has the potential to improve their cost-effective analysis to ameliorate the design of policy interventions that target pressures, thus improving water status and, at the same time, contributing towards increasing the benefits societies obtain from the environment [29].

In this study, we developed accounts of the asset value of two ecosystem services in two areas in Europe that are managed under the Water Framework Directive. To do this, we utilized the ecosystem services concept and the principles of the natural capital methodology. We showed that the data included in the WFD River Basin Management Plans, combined with national statistics, could potentially be used to assess the value of the flow of benefits from efficiently managed water resources. Our aim was to explore the benefits of such an approach in a country that has institutionalized it and in a country that has not yet started the process of developing environmental accounts. The estimation of the stock value of ecosystem services in the UK case study was relatively easy, as national databases and databases containing background information of the River Basin Management Plans were publicly available. On the contrary, in the case of Evrotas, data besides that found in the River Basin Management Plan was limited. As a result, a more sophisticated technique was used to estimate the stock value of recreation in the Broadland Rivers case, which provides greater confidence in the obtained values.

While a discussion on the suitability of PoMs is out of the scope of this study and cannot be supported by processed data, we suggest that nature-based solutions might be more appropriate for increasing the benefits obtained from the environment, while benefiting the environment at the same time. Green infrastructure [161], another name for nature-based solutions have the potential to make the implementation of overlapping policies and legislation more efficient [35] and also generate a high number of co-benefits to society (e.g., enhancement of riverbank vegetation for managing erosion also generates benefits in the form of carbon sequestration). Such policy interventions go beyond managing nature effectively by focusing on societal factors, such as human well-being and poverty alleviation and development while sustaining or improving environmental conditions. Eggermont et al. [162] classify three types of nature-based solutions according to the degree of technical intervention: i) Better use of ecosystem through minimal interventions; ii) Approaches that relate to the development of sustainable and multifunctional ecosystems; iii) Creation and management of new ecosystems. Maes and Jacobs [163] define nature-based solutions as "any transition to a use of ecosystem services with decreased input of non-renewable natural capital and increased investment in renewable natural processes". For example, wetland and floodplain restoration are attractive options, as they offer a high degree of risk protection, have the potential to provide ecosystem services benefits beyond the scope of intervention, and are less costly compared to grey infrastructure alternatives [164]. Assuming that such measures could achieve the primary objective of the WFD, nature-based solutions could assist in maximizing the benefits associated with better conditions of water resources, which could effectively increase the value of natural capital. Nevertheless, claims concerning the relationship between different types of PoMs and natural capital should be further investigated.

Finally, a shortcoming of the study is that it focused on two ecosystem services rather than the whole spectrum of benefits provided by the rivers in the two catchments. Data constraint was the primary reason for this choice. Though, to our knowledge, the current study is the first

that shows how data from River Basin Management Plans can be used for assessing the value of natural capital, though further development of the national databases containing environmental information is needed to obtain better results. More specifically, casting light on the relationship of nature and society requires time series data on various social and economic aspects to be gathered in fixed intervals, for example, per every one or two years. That is particularly relevant for the Greek case, where concise databases do not exist. As a result, further investing in the creation of such repositories of information is required, along with the establishment of common protocols for data collection. The WFD, along with other environmental Directives and EU policies, provide a solid base with which to define the collected data needed to support transdisciplinary management practices and the adoption of holistic frameworks.

**Author Contributions:** Conceptualization, I.S. and N.V.; methodology, I.S. and N.V.; formal analysis, I.S. and N.V.; investigation, I.S. and N.V.; resources, I.S. and N.V; data curation, I.S.; writing—original draft, I.S. and N.V., writing—review and editing, I.S. and N.V.; visualization, I.S. and N.V.; supervision, N.V. All authors have read and agreed to the published version of the manuscript.

**Funding:** This research received no external funding.

**Institutional Review Board Statement:** Not applicable

**Informed Consent Statement:** Not applicable

**Data Availability Statement:** The data presented in this study are available upon request from the corresponding author. The data are not publicly available due to a conditional request from the source.

**Conflicts of Interest:** The authors declare no conflict of interest.

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
