# Peer review of "Natural Capital Accounting Informing Water Management Policies in Europe"

_sustainability, doi:10.3390/su132011205_

Round 1
Reviewer 1 Report
General comments:
The paper presents a natural capital accounting methodology to estimate the asset value of ecosystems services and applied this methodology in two river basins in Europe. The paper is well-written and presented. It is well referenced.
However, it is not clear to me how the estimation of the costs is done. For example, table 1 shows the value of water for residential use in Evrotal case study, taking into account the number of residents, the assumed water consumption value of 250 L per resident per day and "the pricing scale and capital and operating costs of the supplier". How much are they? What are the original data used? Where are they taken from? the suppliers or the River Basin Plan? The same stands for the Broadland rivers case study.
Another important aspect is that only the residential water demand is taken into consideration. The authors state that there are not enough data in order to take into consideration other water uses. However, residential water demand is usually a small portion of the total water demand (in most cases, and especially in the Mediterranean countries, agricultural water demand is the dominant one). How is this taken into consideration in your study? Do you think that the results will be different when other water demands are used for the estimation of the annual value? If yes, how?
Lines 553-570: Is the water availability index taken into consideration? The authors arrive to conclusions taking into consideration the values estimated. However there is no discussion on other related environmental indices.
What about reliability of the data?
In general, I find the authors' study very interesting and significant. However at least some important aspects need to be discussed, if not taken into account. Also, the methodology should be more clear to the readers.
Specific comments:
- the authors can enhance the abstract section with the most important results from their study
- please correct all words thoughout the manuscript separated with a slush (-), e.g. line 39, "cur-rent"
- After conclusions section and before references section, the authors should add other sections as presented in the template, for example author contributions, funding, data availability statement, etc.
Author Response
- However, it is not clear to me how the estimation of the costs is done. For example, table 1 shows the value of water for residential use in Evrotal case study, taking into account the number of residents, the assumed water consumption value of 250 L per resident per day and "the pricing scale and capital and operating costs of the supplier". How much are they? What are the original data used? Where are they taken from? the suppliers or the River Basin Plan? The same stands for the Broadland rivers case study.
This section has been revised to better explain the process of computing the value of water for residential purposes in both cases. Additionally, a footnote has been included to provide information on the computed values as requested by the reviewer.
- Another important aspect is that only the residential water demand is taken into consideration. The authors state that there are not enough data in order to take into consideration other water uses. However, residential water demand is usually a small portion of the total water demand (in most cases, and especially in the Mediterranean countries, agricultural water demand is the dominant one). How is this taken into consideration in your study? Do you think that the results will be different when other water demands are used for the estimation of the annual value? If yes, how?
The paper presents how the stock value of two ecosystem services was obtained in two different case studies. Without a doubt in order to estimate the total value of water, one would need to assess the full range of ecosystem services and estimate their value. We focused on two categories of ecosystem services as the aim of the study was to showcase how the natural capital accounting methodology can be applied using existing WFD data. Undoubtedly, existing information would not suffice to estimating the asset value in every catchment in Europe. This is why we mention that in order for Member States to use natural capital accounting in policy making they are required to invest in collecting a wide range of information. Estimating the value of other water uses would in fact be interesting, as results could be drawn on the overall use of water. Nevertheless, our results are not affected by not assessing the value of other ecosystem services, as the value of each ecosystem service could be evaluated separately.
- Lines 553-570: Is the water availability index taken into consideration? The authors arrive to conclusions taking into consideration the values estimated. However there is no discussion on other related environmental indices.
That comment was particularly useful. We revised the discussion to mention the complementarity of natural capital accounting and environmental indices in policy making.
- What about reliability of the data?
We included comments about that in the revised version of the paper. Overall, we consider the data to be reliable as it undergoes the scrutiny of the national authorities and the European Commission.
- In general, I find the authors' study very interesting and significant. However at least some important aspects need to be discussed, if not taken into account. Also, the methodology should be more clear to the readers.
We believe that the revised version presents the methodology in a clearer way.
Specific comments:
- the authors can enhance the abstract section with the most important results from their study
The abstract has been revised.
- please correct all words thoughout the manuscript separated with a slush (-), e.g. line 39, "cur-rent"
This was a problem inherited by the used template, but it has been corrected.
- After conclusions section and before references section, the authors should add other sections as presented in the template example author contributions, funding, data availability statement, etc.
The additional sections have been included.
Reviewer 2 Report
The topic is outstandingly topical and fit to the journal themes. The paper brings new information for readers about effort of valuation of natural capital with very detailed literature review and also has contribution to the solving some issues of problems based on the demonstration of possible application of authors idea in two catchments in Europe.
The paper is scientific level, but there are places for improvements.
In the abstract: Using primary and secondary methods, the asset value of two ecosystem services is estimated and associated with changes in water status due to policy instruments.
Not really understandable for me, it is needed to explain what is the meaning of primary and secondary method and what kind of ecosystem services the authors talking about.
The following statement is too general and wellknown. “Findings demonstrate that as the asset value of water resources is influenced by both current and future uses, managing authorities should consider both current and emerging pressures when designing programmes of measures to manage water resources sustainably.” The abstract reversion is needed.
Introduction is well written but the source of figures are missing. That is cited or own construction based on literature?
The introduction is too long. I suggest to follow the usual structure of scientific papers, Literature review following the introduction. At the end of introduction please explain the structure of paper.
At the end of literature review it would be nice make clear, how the own research related to them, what from presented methodology will be used. At the end of the Literature review part, I miss a formulation of research hypotheses that would be connected with the previously published studies and also give information about the potential value added of paper.
In the subchapter “Estimation of the value of natural capital ” The author should inform the readers, what kind of the natural capital is examined, and make a list of factors effect on the value of that.
In the subchapter “Estimation of the value of natural capital ” The author should inform the readers, what kind of the natural capital is examined, and make a list of factors effect on the value of that.
In the discussion part the author should make much more clear, how their results and statement of literature cited here are related.
The Conclusion part seems to be a summary and some parts are related to literature review.
The statement and suggestion of authors should be pointed out and also the potential implications should be more highlighted in this part.
Author Response
- In the abstract: Using primary and secondary methods, the asset value of two ecosystem services is estimated and associated with changes in water status due to policy instruments. Not really understandable for me, it is needed to explain what is the meaning of primary and secondary method and what kind of ecosystem services the authors talking about.
- The following statement is too general and wellknown. “Findings demonstrate that as the asset value of water resources is influenced by both current and future uses, managing authorities should consider both current and emerging pressures when designing programmes of measures to manage water resources sustainably.” The abstract reversion is needed.
The abstract has been revised.
- Introduction is well written but the source of figures are missing. That is cited or own construction based on literature? The introduction is too long. I suggest to follow the usual structure of scientific papers, Literature review following the introduction. At the end of introduction please explain the structure of paper. At the end of literature review it would be nice make clear, how the own research related to them, what from presented methodology will be used. At the end of the Literature review part, I miss a formulation of research hypotheses that would be connected with the previously published studies and also give information about the potential value added of paper.
While we have not made major changes in this part as the topic has not been analyzed in the past and we wanted to make sure that it is well-framed, we believe that given the current version of the article is more structured for the reader. Regarding the figures, as they were developed by the authors, so no sources had to be included.
- In the subchapter “Estimation of the value of natural capital ” The author should inform the readers, what kind of the natural capital is examined, and make a list of factors effect on the value of that.
Introductory text has been included to better present the subsequent sections.
- In the discussion part the author should make much more clear, how their results and statement of literature cited here are related.
The part has been revised.
- The Conclusion part seems to be a summary and some parts are related to literature review. The statement and suggestion of authors should be pointed out and also the potential implications should be more highlighted in this part.
This part has also been revised to better express what the study suggests.
Round 2
Reviewer 1 Report
The revised version of the paper addresses all my comments. The paper is very interesting, well-written and presented. Thus it deserves publication and it can be accepted for publication at Sustainability journal.
Reviewer 2 Report
All my suggestion are taken into consideration or explained the authors' consept on acceptable way.